# Application of Immersive Virtual-Reality-Based Puzzle Games in Elderly Patients with Post-Stroke Cognitive Impairment: A Pilot Study

**DOI:** 10.3390/brainsci13010079

**Published:** 2022-12-31

**Authors:** Zhilan Liu, Zhijie He, Jing Yuan, Hua Lin, Conghui Fu, Yan Zhang, Nian Wang, Guo Li, Jing Bu, Mei Chen, Jie Jia

**Affiliations:** 1Department of Rehabilitation Medicine, Huashan Hospital, Fudan University, Shanghai 200040, China; 2Department of Geriatric Rehabilitation Medicine, Shanghai Fourth Rehabilitation Hospital, Shanghai 200040, China; 3Shanghai Jinshan Zhongren Aged Care Hospital, Shanghai 201501, China; 4National Clinical Research Center for Aging and Medicine, Huashan Hospital, Fudan University, Shanghai 200040, China; 5Department of Rehabilitation Medicine, Shanghai Jing’an District Central Hospital, Shanghai 200040, China; 6National Center for Neurological Disorders, Shanghai 200040, China

**Keywords:** immersive virtual reality, stroke, post-stroke cognitive impairment, aging

## Abstract

Background: The society is aging in China, and the cognitive level of elderly post-stroke patients gradually declines. Face-to-face cognitive functional training is no longer sufficient. Immersive virtual reality (IVR) is a promising rehabilitation training device. In this study, we developed an IVR-based puzzle game to explore its effectiveness, feasibility, and safety in elderly stroke patients with cognitive dysfunction. Methods: A total of 30 patients with mild post-stroke cognitive impairment after stroke were randomly assigned to a control or IVR group. Patients in both groups received routine rehabilitation therapy. Patients in the control group received traditional cognitive training, and those in the IVR group received IVR-based puzzle game therapy. Before and after treatment, Montreal cognitive assessment (MOCA), trail-making test-A (TMT-A), digit symbol substitution test (DSST), digital span test (DST), verbal fluency test (VFT), and modified Barthel index (MBI) were evaluated in both groups. In addition, the IVR group was administered a self-report questionnaire to obtain feedback on user experience. Results: There was no significant difference in the baseline data between the two groups. After six weeks of treatment, the cognitive assessment scores were improved in both groups. Moreover, the IVR group showed more improvements than the control group in the DSST (Z = 2.203, *p* = 0.028 < 0.05, *η*^2^ = 0.16); MOCA (T = 1.186, *p* = 0.246 > 0.05, d = 0.44), TMT-A (T = 1.791, *p* = 0.084 > 0.05, d = 0.65), MBI (T = 0.783, *p* = 0.44 > 0.05, d = 0.28), FDST (Z = 0.78, *p* = 0.435 > 0.05, *η*^2^ = 0.02), BDST (Z = 0.347, *p* = 0.728 > 0.05, *η*^2^ = 0.004), and VFT(Z = 1.087, *p* = 0.277 > 0.05, *η*^2^ = 0.039) did not significantly improve. The significant difference in DSST represents an improvement in executive function and visual–spatial cognitive characteristics. The other assessment scores did not show such features. Therefore, we did not observe significant differences through this measure. According to the results of the self-report questionnaire, most of the patients were satisfied with the equipment stability and training content. Several individuals reported mild adverse reactions. Conclusions: This pilot study suggests that IVR-based puzzle games are a promising approach to improve post-stroke cognitive function, especially executive cognitive function, and visual–spatial attention in older adults.

## 1. Introduction

The population aging is proceeding at an accelerating rate in China. In 2020, the number of people over 65 years old was 1906.4 million, accounting for 13.5% of the total population [1]. The number of elderly people who are unable to live independently and need help due to cognitive decline and dementia has also relatively increased. Stroke is the leading cause of long-term physical and cognitive impairment and death in China. Cognitive degeneration is significantly faster in elderly patients after stroke than in healthy elderly people [2]. More and more studies are showing that the cognitive function of patients with stroke sequelae will continue to decline [3,4], including global cognition, attention, processing speed, memory, language, perceptual motor skills, and executive function [2,5]. The prevalence of dementia ranges from 7% to 49.8% 1 to 10 years after stroke [3,6,7]. Cognitive impairment not only affects the lives of individuals, caregivers, and families, but also places a heavy economic burden on medical resources. It has become one of the main causes of dysfunction, institutionalization, and death of the elderly in China.

Cognitive training has been proven to improve mild cognitive impairment after stroke [8,9]. The traditional cognitive training pattern usually uses paper and tools, requiring the therapist to interact face-to-face with the patient, and is limited by therapeutic tools that cannot meet the treatment needs. After long-term training, patients feel bored and lose their enthusiasm for training, so their compliance with treatment reduces. With the development of science and technology, cognitive training is no longer limited to the therapist’s manual interventions. The elderly are increasingly aware and accepting of computer technology due to the pervasiveness the Internet, computers, mobile phones, and other electronic devices in daily life.

Compared with traditional cognitive training methods, immersive virtual reality (IVR) is an intelligent technology that uses a head-mounted device (HMD) or cave automatic virtual environment (CAVE) equipped with motion sensors to artificially create a virtual environment similar to the real world. Because it can present three-dimensional objects and create complex visual, auditory, and tactile multisensory virtual environments, it brings people an immersive feeling of daily activities, so is more and more popular in the medical and rehabilitation fields [10]. This nonintrusive multisource stimulation can more effectively improve the level of nerve remodeling in damaged brain regions, which is conducive to rehabilitation therapy [11].

Some studies have applied IVR to improve mood [12,13], balance gait [14,15,16], or limb function [17] in the elderly population; however, there is limited academic evidence on how the elderly after a stroke with cognitive impairment experience IVR [18]. Most studies have focused on cognitive dysfunction in the acute phase [17,18,19], but ignore cognitive decline in the chronic phase. On the other hand, the training content and scene design of IVR are relatively unitary. For example, Huang et al. [20] designed IVR recall scenarios to improve and maintain cognitive function in elderly patients with dementia. Researchers from South Korea improved attention and executive function in patients with MCI by designing a 3D VR program for shopping at a supermarket [21]. Manera et al. [22] tried to develop a kitchen and cooking game to assess and rehabilitate elderly people with MCI and Alzheimer’s. Currently, there are still no systematic and comprehensive cognitive training programs for IVR applications on the market.

To solve these problems, we combined IVR with puzzle games. Puzzle games are patterns with different forms and content and a variety of training programs. Recent studies have indicated that puzzle games are enjoyable; repeatable; easy to operate by the elderly; and can improve attention, visuospatial, and executive functions [23,24,25]. Compared with the literature, the advantages of our method are: the IVR puzzle game system contains plentiful training content, which contains 3 major categories and 16 intelligible puzzle game items from which therapists and patients can choose. Our IVR uses a light HMD, stress-free smart sensor. It is easy for the elderly to understand and operate. Completion of the task is not dependent on motor function, and the proposed IVR is suitable for patients with varying degrees of limb impairment. It can also be combined with rehabilitation equipment during training, which is practical. The game interface is clear and real, producing a strong feeling of “faking the real”. The system has an automatic feedback system and does not rely on human supervision. We selected elderly patients in the chronic phase after stroke and aimed to explore the initial effectiveness, feasibility and safety of this intelligent training system in elderly patients with post-stroke cognitive impairment.

## 2. Materials and Methods

### 2.1. Participants

This study was performed from January 2022 to October 2022. A total of 30 elderly patients with post-stroke cognitive impairment were recruited from the Department of Geriatric Rehabilitation, Shanghai Fourth Rehabilitation Hospital. The inclusion criteria were as follows: (1) stroke was diagnosed according to the criteria of Chinese Guidelines for the Prevention and Treatment of Cerebrovascular Diseases; (2) 60 ≤ age < 90 years old, with stroke onset ≥ 6 months; (3) met the diagnostic criteria of PSCI [26]; (4) Montreal cognitive assessment scale (MoCA): 18 ≤ MoCA < 26; (5) Fugl–Meyer motor scale > 85 for at least one upper and lower limb; (6) educational attainment ≥ 9 years. The exclusion criteria were as follows: (1) those who were difficult to evaluate or examine or could not cooperate with instructions; (2) patients with severe hearing impairment, visual impairment, mental disorders, or a history of epilepsy; (3) patients with previous vertigo; (4) those who were participating in other clinical studies of cognitive function at the same time. The discontinuation indicators were as follows: (1) patients who were unwilling to continue the trial; (2) intolerable or serious adverse events occurred, such as severe dizziness, vertigo, or other discomforts, and the investigator judged that the risk to the patients was greater than that to the beneficiaries; (3) during the study period, the patient’s condition deteriorated or had a recurrent stroke; (4) unplanned discharge of the patient.

All individuals signed an informed consent form before the start of the study. The study was approved by the Ethics Committee of Huashan Hospital (KY2022-549) and registered with the Chinese Clinical Trial Registry (ChiCTR2200061932)

### 2.2. Study Design

In our study, PASS15 software (NCSS LLC., Kaysville, UT, USA) was used to calculate the sample size based on the assumption of equal variance of two samples. We set the power to 0.80 and α to 0.05. The Montreal cognitive assessment (MoCA) was used as the outcome index according to previous similar studies [27]. We considered the cognitive improvement effect of the IVR group was 3.2 higher than that of the non-IVR group as effective, the standard deviation of the individual MoCA was 2.2, and the loss rate was 20%. Finally, 15 cases in the experimental group and 15 cases in the control group were enrolled, for a total of 30 cases.

The 30 individuals were randomly divided into a control group (CG) or immersive virtual reality group (IVRG); each group has 15 individuals. All individuals received routine secondary medicine stroke prevention and 90 min of daily rehabilitation physiotherapy and occupational therapy. After that, individuals in the CG received traditional cognitive training and, those in the IVRG received IVR training. Traditional cognitive training included (1) processing speed and attention training: Schulte table training; (2) memory training: retelling content after seeing pictures such as cards and calendars; (3) computational ability training: performing addition and subtraction operations within 100; (4) executive and problem-solving ability training: such as using building block shapes, picture information classification, and reasoning simulation training. The training content of the IVRG system includes 3 categories: life skills training, exergames and entertaining games; a total of 16 game items are included (Figure 1). The difficulty level of each game is divided into five stars, where one star is the simplest and five stars is the most difficult. Investigators selected one item in each category in turn according to the patient’s interest. Individuals in the IVRG wore head-mounted displays for training and started at the difficulty level of one star. The difficulty of the training was gradually adjusted from simple to complex, and each item lasted 5 min, with 2 min rest between the items. During the treatment, if individuals experienced any intolerable discomfort, the treatment was immediately stopped. The extra intervention time was the same in both groups: 15 min per day, 6 sessions per week, for a total of 6 weeks. Cognitive function was assessed for all subjects before and after 6 weeks of treatment, and self-report questionnaires were administered only for the IVR group after 6 weeks of training. Assessors were therapists who were specifically trained but not involved in the intervention study. The assessment was conducted in a quiet room in the Department of Rehabilitation Assessment. The evaluator was responsible for the evaluation, collecting the evaluation and questionnaire results of all subjects, and making the final statistical analysis.

### 2.3. Assessments and Questionnaire

The global cognitive function was used as the primary outcome, and the MOCA-Beijing version was used for evaluation [28]. Each score was added to the final score if the patient’s education years was ≤ 12 years plus 1 point, and the highest score was 30 points [29].

Secondary indicators include the trail-making test-A (TMT-A) [30], which assesses attention and information processing speed; the digit symbol substitution test (DSST) [31], which is used to measure executive function and visuospatial attention; the digital span test (DST) from the Wechsler adult intelligence scale IV in China [32], which is used to evaluate performance and memory; verbal fluency test (VFT), animal category [33], which used to assess language ability; and modified Barthel index (MBI) [34], which is used to assess activities of daily living.

TMT has good objectivity and is recommended by the National Institutes of Neurological Disorders and Stroke–Canadian Stroke Network (NINDS-CSN) as one of the three scales to evaluate executive function [35]. DSST is relatively independent of intelligence, education, and age, and is suitable for the elderly [31]. DST includes the forward digit span test (FDST) and backward digit span test (BDST). There are 10-digit strings in the FDST and 9-digit strings in the BDST. They measure immediate memory or short-term memory, attention, and working memory [32]. VFT is a simple measure of semantic fluency that requires individuals to name as many animals as possible in one minute and is widely used for cognitive assessment in older adults [36]. MBI is one of the most recommended methods for measuring activities of daily living, having good internal consistency, and is suitable for stroke patients [37].

The self-report questionnaire mainly consisted of 3 parts and 14 items to investigate the IVR group (Appendix A). Part I inquired how often individuals used smart devices before the intervention. Part II was the satisfaction survey regarding IVR equipment and training content after 6 weeks of IVR intervention using a five-point Likert scale (ranging from “strongly disagree”, designated as 1, to “strongly agree”, designated as 5). In Part III, the visual analogue scale (VAS) was used to record the occurrence and degree of adverse reactions during the intervention.

### 2.4. VR Equipment

#### 2.4.1. VR Integrity System Construction

Our study used a virtual reality post-stroke intelligent motor training system that was divided into 4 modules, including observation, doctor, patient, and server terminals. The observation and doctor terminals were installed on the same Android tablet. The doctor terminal, as a signal source for IVR all-in-one networking, was responsible for managing the patient’s personal information and hardware equipment information and controlling the start and stop of the patient’s rehabilitation. For the patient terminal, patients wore the HMD. The direction of motion in the HMD screen could be changed by turning the head. The sensor used 3DOF inertial motion technology and a built-in acceleration gyroscope. Wearing the smart sensor on the patient’s limb or motion device, the angle and acceleration could be fed back to the patient to complete the training. The observation terminal was connected via Bluetooth, a sensor, and VR all-in-one, allowing the doctor to watch the scene of patient training in real time on the Android tablet, and to perform hand position, head position correction, and horizontal switching while guiding patient training. The architecture used by the server terminal was a Spring MVC, which was a module for web development based on the Spring framework, and the database was MYSQL5.6 established by MySQL AB. The system could save the patient’s training data at any time, which were fed back to the doctor after processing (Figure 2).

#### 2.4.2. Scene Modeling

The aim of scene modeling is to be as close to reality as possible based on interest. Therefore, as the most critical and directly touched limb in the scene, 1:1 modeling was used to restore the human arm function to the greatest extent possible. In addition, considering the acceptance of the virtual environment by the elderly, we created a comfortable environment to relax the individuals. For the game background, we selected quiet and relaxed venues such as beaches, street scenes, and parks. Background music was melodious, soothing, and calm. For the construction of the visual environment, we used 8K high-definition 360 degree panoramic shooting to create the most realistic scene.

### 2.5. Statistical Analysis

SPSS version 20.0 (IBM Inc., Chicago, IL, USA) was used for statistical analysis. Measurement data were tested for normal distribution and homogeneity variance. Enumeration data were analyzed by Fisher’s precision probability test. Normally distributed data are expressed as mean ± standard deviation, a paired-sample t-test was used for intragroup comparison, and the independent-sample t-test was used for intergroup comparison. Skewed distributions are expressed as the median (interquartile range), the Wilcoxon W test was used for intragroup comparison, and the Mann–Whitney U test was used for intergroup comparison. We considered differences with a two-sided *p* < 0.05 to be statistically significant. Self-report questionnaires were analyzed as percentages.

## 3. Results

### 3.1. Baseline Clinical Data

A total of 30 patients were included in our study, with an average age of 74.16 ± 7.08 years old, with 15 cases in the IVRG, and 15 cases in the CG. All patients completed the training, including 17 men and 13 women, 21 cases of cerebral infarction, and 9 cases of cerebral hemorrhage. There was no significant difference in the baseline data between the two groups (*p* > 0.05) (Table 1).

### 3.2. Results of the Cognitive Evaluation

There were no significant differences in MOCA, TMT-A, MBI, DSST, FDST, BDST, or VFT scores between the IVR and control groups before treatment (*p* > 0.05) (Table 2). After 6 weeks of treatment, the scores for the MOCA (IVRG: T = 8.981, *p* < 0.001; CG: T = 13.229, *p* < 0.001), TMT-A (IVRG: T = 5.644, *p* < 0.001; CG: T = 4.112, *p* = 0.001), MBI (IVRG: T = −2.779, *p* = 0.015; CG: T = −6.089, *p* = 0.000) (Figure 3), DSST (IVGR: Z = 3.422, *p* = 0.001; CG: Z = 3.482, *p* < 0.001), FDST (IVGR: Z = 2.887, *p* = 0.004; CG: Z = 2.121, *p* = 0.034), BDST (IVGR: Z = 3.317, *p* = 0.001; CG: Z = 2.111, *p* = 0.035), and VFT (IVGR: Z = 3.332, *p* = 0.001; CG: Z = 3.429, *p* = 0.001) (Figure 4) in both groups significantly improved compared with those before treatment (*p* < 0.05). The comparison of scores after treatment between the two groups showed that the DSST scores of the IVR group (21,6) were higher than those of control group (14,11), and the difference was statistically significant (Z = 2.203, *p* = 0.028 < 0.05, *η*^2^ = 0.16); MOCA (T = 1.186, *p* = 0.246 > 0.05, d = 0.44), TMT-A (T = 1.791, *p* = 0.084 > 0.05, d = 0.65), MBI (T = 0.783, *p* = 0.44 > 0.05, d = 0.28) (Figure 3), FDST (Z = 0.78, *p* = 0.435 > 0.05, *η*^2^ = 0.02), BDST (Z = 0.347, *p* = 0.728 > 0.05, *η*^2^ = 0.004), and VFT (Z = 1.087, *p* = 0.277 > 0.05, *η*^2^ = 0.039) were not significantly improved. The size effect of the differences between groups in DSST was *η^2^* = 0.16 > 0.14 (*p* = 0.028 < 0.05), which means the significant difference in DSST was reliable. 

### 3.3. Self-Report Questionnaire

In the IVRG, 73.33% of the elderly stroke patients had never used smartphones. They used geriatric cellular phones with buttons or had no phones. A total of 60% of individuals had never used portable Android devices (PADs), and 20–13.33% of individuals seldom or sometimes used PADs to watch TV shows or play games. Of the individuals, 60% were still willing to use smart devices and 66.67% could completely understand the training directive. None of the individuals were dissatisfied with the training content, and 53.34% of them derived enjoyment from the IVR training process. More than 73.33% of individuals approved of the stability and maneuverability of our IVR device. A total of 73.34% of individuals were willing to promote the IVR device (Figure 5).

### 3.4. Side Effects

During IVR intervention, there were two individuals who reported dizziness without nausea and vomiting, and their VAS score was one. Two individuals reported dry eyes, and three individuals reported eye fatigue, of which two individuals had a VAS score of one and one individual had a VAS score of two (Figure 6). The other individuals did not report any adverse reactions that occurred.

## 4. Discussion

The purpose of this study was to investigate the initial effect and feasibility of IVR-based puzzle games in elderly patients with cognitive impairments after stroke. We initially found that IVR could improve their cognitive function, especially the executive functioning and visuospatial attention of elderly stroke patients. It is feasible to provide them with 15 min of IVR training 6 times a week.

### 4.1. Effectiveness of IVR-Based Puzzle Games

After six weeks of treatment, the global cognitive function (MoCA), attention and information processing speed (TMT-A), executive function and visuospatial attention (DSST), performance and memory (FDST and BDST), language ability (VFT), and activities of daily living (MBI) in both groups significantly improved. Moreover, the IVRG showed more improvements than the CG in the DSST (Z = 2.203, *p* = 0.028 < 0.05). This result is similar to those of a previous meta-analysis [38] of 894 patients from 23 randomized controlled trials, which showed significant improvements in executive and visuospatial function after VR interventions compared with conventional rehabilitation. However, there were no significant differences observed in global cognitive function, attention, verbal fluency, depression, or quality of life.

The significant difference in DSST represents an improvement in executive function and visual–spatial cognitive characteristics. The other assessment scores such as DST and MBI, did not show such improvements. Therefore, we did not observe significant differences in this measure. The DSST test required subjects to fill in the corresponding symbols of numbers on a blank form as fastest as possible within 90 s. In this process, the subjects needed to match nine digits corresponding to symbols and fill them in within the time limit. Executive functions include task setting, behavioral initiation, monitoring, and self-regulation [39]. In older adults, these domains all play an important role in cognitive tasks. For example, in our preset training, when completing the task of cooking, subjects needed to understand the whole cooking process, from washing vegetables under the tap, cutting dishes, to cooking. In this process, the system broadcasted the steps that needed to be completed by voice prompts, and only when the content of one stage was complete could the subject proceed to the next step, which required the subject to make appropriate choices and respond to the task.

The underlying mechanism through which IVR training improves cognitive function is unclear. The current possible hypothesis is that the virtual environment stimulates and activates brain metabolism, increases cerebral blood flow and neurotransmitter release [40,41], and reactivates and improves various cortical functions [41,42]. A study also found that the input of sensory function activates the brain regions associated with executive function [43]. Virtual reality technology is characterized by immersion, interactivity, and imagination [44]. Participants are isolated from the real world by different sensory capture devices [45]. IVR allows individuals to interact with virtual environments and three-dimensional entities to promptly obtain natural feedback information. The multiple sensory function declines experienced in elderly patients are associated with cognitive decline [46,47]. Especially for elderly patients, multisensory intervention learning is more beneficial than single-sensory intervention for those with cognitive decline [48,49]. Our IVR-based puzzle game system integrates interactive screens, immersive vision, voice prompts, and physical vibrations to stimulate the various senses of the participants in the virtual environment.

Exergames are thought to improve both motor function and cognition in older adults [43], which are defined as physical exercise that combines interaction with cognitive stimulation in a game environment. In our study design, the post-stroke elderly patients in the IVRG could choose a suitable or interesting game from options including playing squash or baseball, using an antiaircraft gun, and gliding. The subjects who chose to play squash or baseball were given a stick or, which they could swing as if they were hitting the ball in a game. The antiaircraft gun and gliding required the subject to use a pedal or limb linkage. A HMD was used as the aiming direction, and the head needed to be moved to aim in the shooting direction. The patient also needed to shake their arm or pedal the bicycle. The faster the speed, the more bullets could be emitted. Our results support previous findings. Htut et al. [50] found that virtual-reality-based game exercise not only improved the balance and muscle endurance of the elderly but also increased the patient’s enthusiasm and improved the global cognitive level on the MOCA more than traditional physical activity. Huang [11] suggested that the combination of immersive VR and exergames enhanced the sense of presence during exercise and had the potential to further improve executive function in older adults after 4 weeks of exergame training. Executive function plays an important role in improving the ability to perform activities daily living. However, unfortunately, the IVRG did not show significant advantages over the non-IVRG in our results. This may be related to the standard of the MBI scale. Although the MBI is reliable and effective, it lacks detailed assessment of the cognitive field and the participation in some social activities [51]. Moreover, our puzzle games are extensive but goal-oriented, so need further improvement. The complexity of future life skills training projects and steps need to be further upgraded.

IVR may improve the visual–spatial function of elderly patients through the characteristics of ecological validity [52]. IVR may mistakenly make subjects believe that they are in the real world through immersive stimulation. The interference from the outside world can be eliminated, and the participants can be immersed in the virtual world through increased attention and reduced distraction [19]. In addition, in our study, for the VR background, we selected diversified and comfortable natural environments, and the kitchen, road, and room were all three-dimensional simulated spaces that accurately mimicked real environments, which could improve the patient‘s visual spatial ability through visual stimulation. Kim et al. [53] mentioned in their discussion that for their virtual background, they selected natural landscapes such as mountains, fields, ski resorts, and football fields. Visual attention and short-term visuospatial memory in the VR group were significantly improved in acute stroke patients with cognitive impairment, which was similar to our finding.

IVR can not only be used to train executive ability, spatial disorientation, but also to improve episodic and verbal memory, attention [52], and living ability [18,54]. However, because different training items have varying training effects, the results of our study are different from those of previous studies. Gamito et al. [55] used several activities of daily living as VR cognitive function training content, such as purchasing items, finding routes, finding characters, recognizing signs, and calculating. After 4–6 weeks of two to three sessions per week, stroke patients had improved memory and attention but not visual function. Previous studies have been conducted to help patients improve episodic memory by showing them familiar environments, including streets, residences, and childhood scenes [56,57]. The task of listing supermarket shopping items has also been used to help elderly patients improve age-related memory decline [21,58]. Combined with our research, most of our training was aimed at the patient completing a task but did not involve memorizing or recalling. However, the performance and memory of the IVRG group were improved compared with those before treatment. The reason may be that VR can improve the memory function of the elderly by enhancing concentration [56,57,58].

Our IVR-based puzzle game training program provided variety and enjoyment. It integrates diversified and plentiful life skills, exergames, and entertainment modules. It could improve cognitive function in terms of episodic memory, verbal memory, attention and daily living ability, and had a significant effect on executive ability and spatial orientation. However, previous studies have also pointed out that [55,59] there is insufficient evidence that the cognitive function of the IVR group improved more than that of the non-IVR group. Therefore, we hope that IVR may maintain and, if possible, improve cognitive function in elderly patients with chronic stroke. There are a large number of elderly patients with stroke sequelae in China. At present, the rehabilitation of patients after stroke is carried out in rehabilitation hospitals. The disadvantages are short hospitalization time, shortage of ward beds, and lack of therapists, so are not suitable for long-term rehabilitation of elderly patients. In the long run, community-based rehabilitation can meet the needs of some patients, but only if there are enough therapists. According to the results, the therapeutic effect of IVR with a smart device is consistent with that of therapists and is acceptable to elderly stroke patients, which makes it possible for IVR to replace manual therapy and have the opportunity to be widely used [60,61].

### 4.2. Feasibility and Safety of IVR in Elderly Stroke Patients

The results of self-report Part I (Figure 5A–C) showed that most elderly patients had no experience with using smart devices, and nearly half of them were unwilling to use smart devices. However, the results of the final survey of IVR are encouraging. Almost all the elderly patients gave positive answers to our training content and the manipulability of the IVR equipment, and they were willing to try this new training method. Morganti [62] also reported that elderly participants were initially unfamiliar with the VR device, but after continuous training, they showed enthusiasm for the rehabilitation exercise. In Part II (Figure 5D–F), more than half of the older participants approved of the content of the puzzle games, indicating that they were receptive to such games. The IVR-based puzzle games attracted these older patients for several reasons. On the one hand, a comfortable visual backdrop and gentle auditory experience attracted the participants and enabled them to quickly immerse in the training environment during the training process. The built-in automatic voice broadcast system of the HMD interactive system encouraged participants when they completed one stage of the task to enhance their self-confidence and desire to make further efforts, which could have increase participants’ motivation to persist in completing the training. On the other hand, the three categories of puzzle games were easily understood by the elderly, and the instructions of the game were acceptable and operable. Participants could have fun while training.

It should be noted that one-quarter of the participants had a neutral attitude toward the game contents, which means that the puzzle games need to be further improved to be more suitable for the interests of the elderly. We will adjust the game contents to include activities such as tai chi, painting, cross-stitch, square dancing, chess, cards, and other puzzle activities in the future.

HMD and smart sensors are linked to VR through a network and Bluetooth. If the network is unstable or the Bluetooth disconnects during patient training, it will affect the stability and control performance of the device, including the inability of the doctor, on the observation side, to keep abreast of the patient’s training. About three-quarters of the participants were satisfied with the device, and the rest were neutral (Figure 5G–H). This may be related to network instability during training as well as the interruption of smart sensor transmission.

Considering that all the elder stroke patients were wearing a HMD and using IVR equipment for the first time, we were not sure how well the patients would accept them, so each training session was no longer than 15 min. Similar to other studies [56,57,58], the incidence of dizziness in elderly stroke patients was very low, which automatically resolved after the end of the intervention. In the beginning, we were concerned about the vision problems of the elderly: whether they could see the pictures on the screen after wearing the HMD or whether those who wore glasses would feel uncomfortable wearing the HMD. The participants were tested and asked whether they could see clearly on the screen before training, and no patients complained about related problems after adjusting the position of the HMD. It is feasible for the elderly who wear glasses to wear a HMD at the same time. One patient reported mild dry eyes and eye strain before the intervention, but the IVR did not significantly worsen his symptoms; in addition, he successfully completed the entire study task. A small number of patients will experience eye fatigue, which can be relieved after rest. Dizziness, nausea, headache, dry eyes, eye strain, and other adverse reactions were not reported in the remaining patients. This means that the vast majority of elderly patients with mild cognitive decline after stroke well-tolerated IVR and were receptive to it.

Reliable, sensitive, and safe training environments are provided by precise control and manipulation in a virtual environment. IVR-based puzzle games have great potential to be used for cognitive intervention in elderly patients with mild cognitive decline after stroke. Moreover, if this interactive technology can be used for social contact, interest activities, and other long-term rehabilitation training goals [63], it may become possible to reduce therapist workload by increasing training initiative [48].

### 4.3. Limitations and Prospects

There are some limitations to this study. The sample size in this study was relatively small, so future studies with larger sample sizes are needed. Moreover, six weeks is too short for patients who need rehabilitation for a long time, and the long-term effects of IVR on cognition are unclear. Cognitive assessment scales were used to evaluate the outcomes in our study, but there was a lack of objective measurement tools. Participants could not be blinded to the trial design, but, fortunately, no participants withdrew from the study. Further studies may address these shortcomings in the future by increasing the sample size; using a longer-duration intervention and follow-up; applying task-related electroencephalograph (EEG), functional near-infrared spectroscopy (fNIRS), and functional magnetic resonance imaging (fMRI) for objective evaluation. Considering that improvements in the cognitive domain are related to training content, it is suggested that systematic training content should also be added to improve memory and performance on command tasks.

## 5. Conclusions

Overall, our research preliminarily demonstrated that IVR-based puzzle games may improve global cognitive, episodic memory, verbal memory, attention, and daily living ability, especially executive ability and spatial orientation, in elderly patients with post-stroke cognitive impairments. This intelligent interactive experience has better applicability in the elderly. The IVR-based puzzle game was well accepted and tolerated in elderly stroke patients and can be recommended for use. Our study changes the traditional two-dimensional training mode to increase the authenticity of the training scene, and brought enjoyment to the elderly patients. Virtual reality technology may have the same efficacy as conventional cognitive rehabilitation; as a noninvasive intervention, it may have the advantages of interest and rich content. It may have application value and development prospects in improving the cognitive function of the elderly with chronic stroke.

## Figures and Tables

**Figure 1 brainsci-13-00079-f001:**
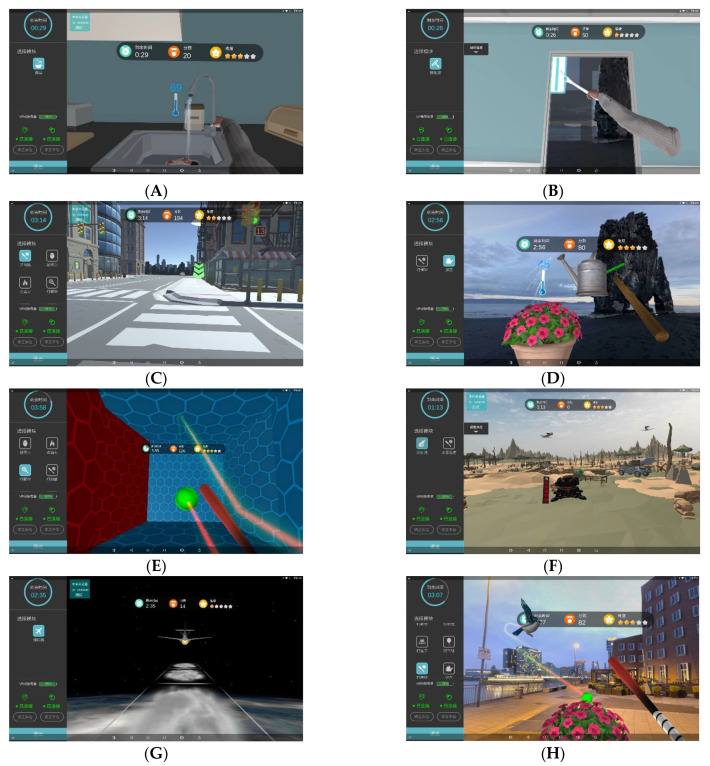
A total of 16 training items were available for researchers and individuals. (**A**–**D**) Life skills: cooking (**A**), cleaning a window (**B**), crossing a road (**C**), and watering flowers (**D**); (**E**–**H**) exergames: playing squash (**E**), shooting antiaircraft guns (**F**), flying gliders (**G**), and playing baseball (**H**); (**I**–**P**) entertaining games: breaking eggshells (**I**), swatting insects (**J**), lighting fireworks (**K**), whack-a-mole (**L**), pumping up a balloon (**M**), flying a Kongming lantern (**N**), Fruit Ninja (**O**), and Bubble Jab (**P**).

**Figure 2 brainsci-13-00079-f002:**
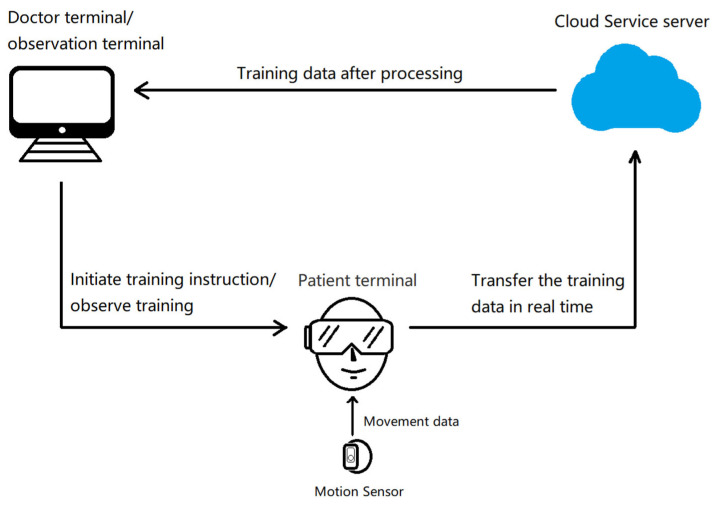
Product structure diagram for immersive virtual reality (IVR).

**Figure 3 brainsci-13-00079-f003:**
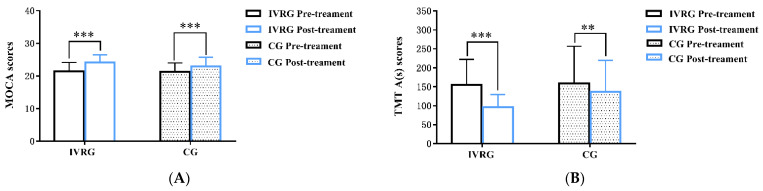
Bar plots for comparison of MOCA (**A**), TMT-A (**B**), and MBI (**C**) between the IVRG and CG before and after treatment. Significant differences were observed in intragroup. *, *p* < 0.05; **, *p* < 0.01; ***, *p* < 0.001. Abbreviation: MOCA, Montreal cognitive assessment; TMT-A, trail-making test-A; MBI, modified Barthel index; IVRG, immersive virtual reality group; CG, control group.

**Figure 4 brainsci-13-00079-f004:**
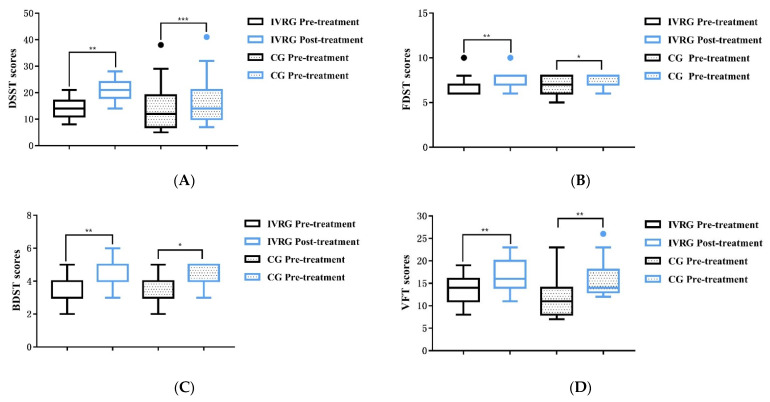
Boxplots comparing DSST (**A**), FDST (**B**), BDST (**C**), and VFT (**D**) between the IVRG and CG before and after treatment. Significant differences were observed in intragroup. *, *p* < 0.05; **, *p* < 0.01; ***, *p* < 0.001. Abbreviation: DSST: digit symbol substitution test; FDST: forward digit span test; BDST: backward digit span test; VFT: verbal fluency test; IVRG, immersive virtual reality group; CG, control group.

**Figure 5 brainsci-13-00079-f005:**
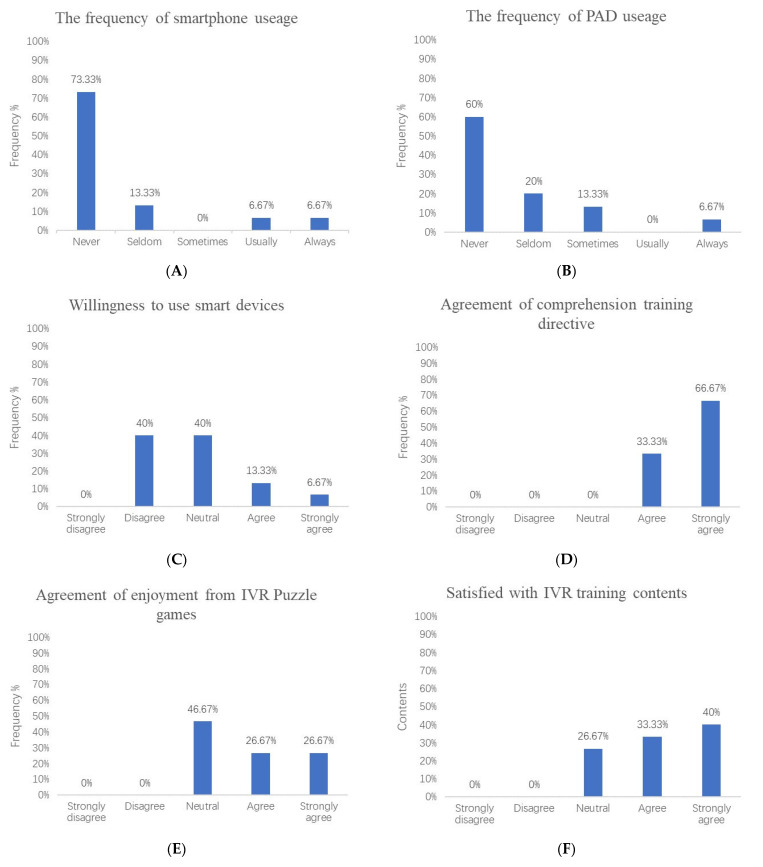
Histograms (**A**–**C**) depict the frequency of smart device usage experienced and the willingness to accept smart devices by the IVR group. Histograms (**D**–**I**) depict satisfaction with IVR equipment and training contents experience. PAD, portable android device; IVR, immersive virtual reality.

**Figure 6 brainsci-13-00079-f006:**
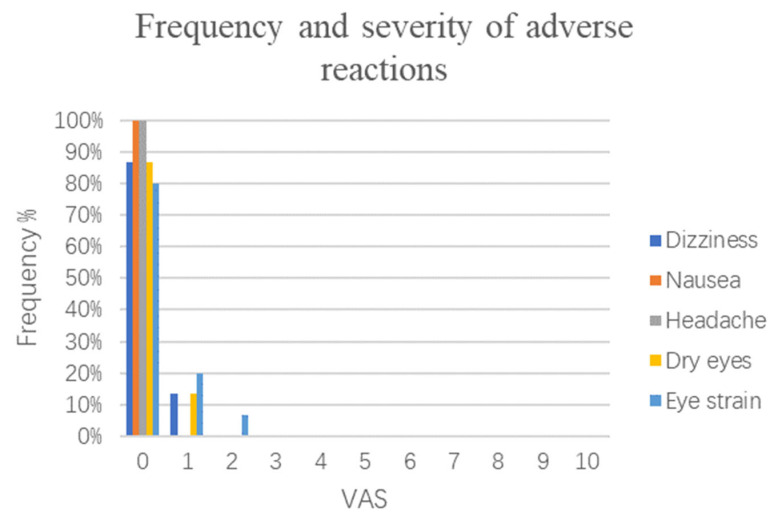
The bar diagram depicts the self-reported incidence and extent of adverse reactions associated with the IVR experience. The horizontal axis represents the percentage of adverse symptoms that occurred, including dizziness, nausea, headache, dry eyes, and eye strain. The vertical axis represents the severity of adverse symptoms experienced by individuals using a VAS, where 0 means no occurrence, and the higher the number, the more severe the adverse symptoms. Five colored lines are used to represent five adverse symptoms. VAS, visual analog scale.

**Table 1 brainsci-13-00079-t001:** Baseline clinical data of elderly stroke patients between the IVRG and CG.

	IVRG *n* = 15	CG *n* = 15	T	*p* Value
Age (Years)	74.93 ± 6.81	73.40 ± 7.5	0.586	0.562
Male/Female	9/6	8/7		1 ^a^
Time since onset (months)	42.93 ± 34.54	29.27 ± 36.51	1.053	0.301
Type, n (%)				0.427 ^a^
Cerebral infarction	12(80)	9(60)		
Cerebral hemorrhage	3(20)	6(40)		

IVRG, immersive virtual reality group; CG, control group; ^a^ Fisher’s precision probability test.

**Table 2 brainsci-13-00079-t002:** Comparisons of baseline cognitive evaluation between the IVR and control groups before treatment.

	IVRG (*n* = 15)	CG (*n* = 15)	T/Z	*p*-Value
MOCA	21.47 ± 2.67	21.27 ± 2.76	0.202	0.842 ^a^
TMT A(s)	155 ± 67.48	159.4 ± 97.33	0.144	0.887 ^a^
MBI	64.67 ± 11.41	59.33 ± 10.83	1.313	0.200 ^a^
DSST	14,6	12,12	0.956	0.339 ^b^
FDST	7,1	7,2	0.441	0.659 ^b^
BDST	4,1	4,1	0.334	0.738 ^b^
VFT	14,5	11,6	1.494	0.135 ^b^

IVRG, immersive virtual reality group; CG, control group; MOCA, Montreal cognitive assessment; TMT-A, trail-making test-A; MBI, modified Barthel index; DSST, digit symbol substitution test; FDST: forward digit span test; BDST: backward digit span test; VFT, verbal fluency test; a, independent-sample t-test; b, Mann–Whitney U test.

## Data Availability

Not applicable.

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
