# Peer review of "Application of Immersive Virtual-Reality-Based Puzzle Games in Elderly Patients with Post-Stroke Cognitive Impairment: A Pilot Study"

_brainsci, 2022, doi:10.3390/brainsci13010079_

Round 1
Reviewer 1 Report
The manuscript describes the results of a neuropsychological rehabilitation in which two groups (VR and control) are compared. Both groups went through a standard training (90min) + 15min of additional training (VR or standard). Both groups improved significantly their performance, with no group difference except for one test, in which VR group improved significantly more. A satisfactory acceptance of the VR technology was recorded.
The manuscript is very well structured and written, with a relatively minor language mistakes.
My main comments/doubts:
1. The VR game is not described with sufficient detail (how long did each exercise last, what was the order, etc.), the images are small and don’t show how the games worked. I believe a video with at least several examples would be very helpful, and the figure with pictures should be moved to the main body of the article.
2. Since both groups improved equally, is it possible that the improvement was a result of the first 90min training, with no added effect of the additional 15min of training (due to fatigue for example)? Authors say it’s worth exploring further this line of treatment, but why spend money and time on building complicated scenarios if it makes no difference in patient’s overall functioning? I feel the argumentation that the majority of patients accepted/enjoyed it is not enough.
3. Do Authors have any explanation for the post-treatment differences in DSST?
4. Please include in the results section the effect sizes.
5. Authors say “There were 66.67% of individuals could completely understand the training directive” – does it mean that over 30% of the sample didn’t understand the instructions?
6. “This means that IVR is more likely to be suitable for the long-term rehabilitation of patients in the chronic phase of stroke…” -> on what ground?
7. “elderly patients are conservative and unwilling to experience new things”-> this seems a (rather harsh) overstatement, especially in the light of the rest of the results
Author Response
Response to Reviewer 1 Comments
Point 1: There were 3 categories of games, each exercise of the IVR game lasted 5 minutes, 2 minutes rest between two items. The total duration of IVR training was 15 minutes, 6 sessions per week for 6 weeks. The training order was Life skills training, Exergames, and Entertaining games. Thank you for your advice. We added an example video to the Supplementary and moved picture to the main body of the article after zoom-in. We have made changes to the above content in the 2.2. Study Design(P3-5).
Response 1: Please provide your response for Point 1. (in red) Point 2: The first 90min training was a routine rehabilitation with no special training on cognitive function. The results were considered to be unaffected by previous studies that did not show improvement in cognitive function with conventional treatment. If a blank control is used, ethical approval may not be granted and the comparability between the two groups is diminished. On the one hand, we hope that IVR may maintain and, if possible, improve cognitive function in elderly patients with chronic stroke. On the other hand, there are a large number of elderly patients with sequelae of stroke in China. At present, the rehabilitation of patients after stroke is carried out in rehabilitation hospitals. The disadvantages are: short hospitalization time, shortage of ward beds, and lack of therapists, which are not suitable for long-term rehabilitation of elderly patients. In the long run, community-based rehabilitation can meet the needs of some patients, but only if there are enough therapists. According to the results, the therapeutic effect of smart device IVR was consistent with that of therapists and is acceptable to elderly stroke patients, which makes it possible for IVR to replace manual therapy and have the opportunity to be widely used
Response 2: Please provide your response for Point 2. (in red)
Point 3: The DSST score is used to measure executive cognitive function and visuo-spatial attention. In older adults, these domains all play an important role in the cognitive tasks. The difference in DSST before and after IVR treatment suggests that VR intervention treatment could significantly improve executive and visuospatial function compared with conventional rehabilitation. The mechanism may be as follows: 1. IVR could stimulate the activation of sensory functions in brain regions related to executive function through multisensory input. 2. We combined exergames to improve executive function in older adults by enhancing presence during exercise and also by increasing patient motivation more than traditional physical activity. 3. IVR may improve the visual spatial of elderly patients through the characteristics of ecological validity and could make subjects mistakenly believe that they are in the real world through immersive stimulation. The interference from the outside world can be eliminated and the participants can be immersed in the virtual world through increased attention and reduced distraction (P11-13 for more details).
Response 3: Please provide your response for Point 3. (in red)
Point 4: 3.2. Results of the Cognitive Evaluation (P7)have made a revision:
There were no significant differences in MOCA, TMT-A, MBI, DSST, FDST, BDST, and VFT scores between the IVR group and control group before treatment (P>0.05) (Table 2). After 6 weeks of treatment, the scores of MOCA(IVRG: T=8.981, p<0.001; CG: T=13.229, p<0.001), TMT-A(IVRG :T=5.644, p<0.001; CG: T=4.112, p=0.001), MBI(IVRG: T=-2.779, p=0.015; CG: T=-6.089, p=0.000)(Figure 2), DSST(IVGR: Z=3.422, p=0.001; CG: Z=3.482, p<0.001), FDST(IVGR: Z=2.887, p=0.004; CG: Z=2.121, p=0.034), BDST(IVGR: Z=3.317, p=0.001; CG: Z=2.111, p=0.035), and VFT(IVGR: Z=3.332, p=0.001; CG: Z=3.429, p=0.001)(Figure 3) in both groups were significantly improved compared with those before treatment(P<0.05). The comparison of scores after treatment between the two groups showed that the DSST scores of IVR group(21,6) were higher than that of control group(14,11), and the difference was statistically significant (Z=2.203, P=0.028<0.05), but there was no significant difference in other assessments scores between the two groups.
Please do not hesitate to contact me again if there is any imperfection in the revision.
Response 4: Please provide your response for Point 4. (in red)
Point 5: We used a five-point Likert scale (ranging from “strongly disagree” designated as 1 to “strongly agree” designated as 5) to investigate the satisfaction survey of IVR. As we can see in Figure 5 (D)(P9) , 66.67% of individuals could completely understand and 33.33% of individuals could understand. These “33.33% of the individuals could understand” means that researchers need to tutor or explain to help patients complete task better.
Response 5: Please provide your response for Point 5. (in red)
Point 6: We hope that IVR may maintain and, if possible, improve cognitive function in elderly patients with chronic stroke (The original article has been revised P12). On the one hand, we selected patients in the chronic phase of stroke, but the definition of long-term is relatively vague. Our study lasted 6 weeks which may not be very long. On the other hand, this is a pilot study, and further studies are needed to demonstrate that IVR improves cognitive function in elderly stroke patients. We hope our study can provide reference for future research.
Response 6: Please provide your response for Point 6. (in red)
Point 7: We agree with the reviewer that this place has been deleted from the original article.
Response 7: Please provide your response for Point 7. (in red)

Reviewer 2 Report
This study aimed to determine the effectiveness, feasibility, and safety in elderly stroke patients with cognitive dysfunction of a rehabilitation intervention, based on an immersive virtual reality (IVR) puzzle game.
The study has potential for clinical applicability, but raises a series of questions regarding the experimental design and the analysis of the obtained results.
First, the group of subjects is too small, and the criteria for selecting study participants are too general. Stroke is a pathology with a clinical and etiopathogenic polymorphism. As a result, subject groups should be constructed based on stricter criteria.
The rehabilitation protocol of the subjects is not clearly defined, such as duration, sequence of application, continuity, intervals between sessions, individualization of the program according to the patient's resources, etc.
The provided data about the evaluation method of the subjects, the sequence of application of the tools, the duration of their application, the predetermined conditions, the training carried out, who administered the questionnaires, etc. are insufficient.
The evaluation tools are not described from the point of view of reliability in clinical practice and the motivation for their choice is not presented.
Regarding the Results section, a problem is that the results are not statistically significant when comparing the control and experimental groups.
If certain results are not statistically significant, concluding is risky. The lack of statistical significance must not be interpreted as a conclusion. Therefore, a finding of no significance means only that judgment should be suspended. Such a conclusion is often a serious misinterpretation because non-significant results are just as often the consequence of insufficient statistical power.
In the abstract, the authors stated that "After six weeks of treatment, the assessment scores were improved in both two groups, but there was no significant difference between the two groups" (lines 32-33). Then, in section 3.2. Results of the Cognitive Evaluation, they added:" The comparison of scores after treatment between the two groups showed that the DSST scores of the IVR group were higher than that of the control group, and the difference was statistically significant (P=0.028), but there was no significant difference in other assessment scores between the two groups" (lines 203-206). There is a contradiction in reporting the results.
Also, for statistically significant results, the effect size should be reported to show how strong the difference between the groups is.
In the Discussion section, authors should discuss the main significant results and how they can be interpreted from the perspective of previous studies and the working hypotheses. The findings and their implications should be discussed in the broadest context possible, with references to recent research findings. Future research directions may also be highlighted.
The conclusions of the study are written ambiguously and do not respect the requirements of research with inferential objectives.
Author Response
Response to Reviewer 2 Comments
1.First, the group of subjects is too small, and the criteria for selecting study participants are too general. Stroke is a pathology with a clinical and etiopathogenic polymorphism. As a result, subject groups should be constructed based on stricter criteria.
Point 1:In our study, the PASS15 software was used to calculate the sample size based on the assumption of equal variance of two samples. We set Power was 0.80, and α was 0.05. The Montreal Cognitive assessment (MoCA) was used as the outcome index according to the previous similar research literature. We assumed that the cognitive improvement effect of the IVR group was 3.2 higher than that of the non-IVR group as effective, the standard deviation of individual MoCA was 2.2, and the loss rate was 20%. Finally, 15 cases in the experimental group and 15 cases in the control group were enrolled, and a total of 30 cases were obtained.
The inclusion criteria we supplemented (2) 60≤aged<90 years old,(5) The Fugl-Meyer motor scale>85 at least one upper and lower limb; (6)educational attainment≥9 years;
Response 1: Please provide your response for Point 1. (in red)
- The rehabilitation protocol of the subjects is not clearly defined, such as duration, sequence of application, continuity, intervals between sessions, individualization of the program according to the patient's resources, etc.
Point 2: All individuals received routine stroke secondary medicine prevention and 90 minutes of daily rehabilitation physiotherapy and occupational therapy. After that, individuals in the CG received traditional cognitive training and IVRG received IVR training. Traditional cognitive training including (1) processing speed and attention training: Schulte table training; (2) memory training: retelling content after seeing pictures such as cards and calendars; (3) computational ability training: doing addition and subtraction operations within 100; (4) executive and problem-solving ability training: such as building block shape, picture information classification, reasoning simulation training. The training contents of the IVRG system include 3 categories: Life skills training, Exergames and Entertaining games and a total of 16 game items are included(Figure 1). The difficulty level of each game is divided into five stars, one star is the simplest and five stars are the most difficult. Investigators selected one item in each category in turns according to the patient's interest. Individuals in the IVRG wore head-mounted displays for training and started with the difficulty of one star. The difficulty of the training was gradually adjusted from simple to complex, and each item lasted 5 minutes, 2 minutes rest between two items. During the treatment, if individuals has any intolerable discomfort, the treatment was stopped immediately. The extra intervention time was the same in both groups: 15 minutes per day, 6 sessions per week for a total of 6 weeks.
Response 2: Please provide your response for Point 2. (in red)
- The provided data about the evaluation method of the subjects, the sequence of application of the tools, the duration of their application, the predetermined conditions, the training carried out, who administered the questionnaires, etc. are insufficient.
Point 3: Cognitive function was assessed for all subjects before and 6 weeks after treatment, and self-report questionnaires were administered only for the IVR group after 6 weeks of training. Assessors were therapists who were specifically trained but not involved in the intervention study. The assessment site was conducted in a quiet room in the Department of Rehabilitation Assessment. The evaluator is responsible for the evaluation, collecting the evaluation and questionnaire results of all subjects and making the final statistical analysis
Response 3: Please provide your response for Point 3. (in red)
4.The evaluation tools are not described from the point of view of reliability in clinical practice and the motivation for their choice is not presented.
Point 4: Thanks for your suggestion, we have added a description of the assessment tool at 2.3. Assessments and questionnaire(P5)
Response 4: Please provide your response for Point 4. (in red)
- Regarding the Results section, a problem is that the results are not statistically significant when comparing the control and experimental groups.
If certain results are not statistically significant, concluding is risky. The lack of statistical significance must not be interpreted as a conclusion. Therefore, a finding of no significance means only that judgment should be suspended. Such a conclusion is often a serious misinterpretation because non-significant results are just as often the consequence of insufficient statistical power.
Point 5: Although there was no significant difference in cognitive function between the IVR group and the non-IVR group after treatment, there was a significant difference in DSST, and there was a significant improvement after treatment compared with before treatment. On this point, in order to express more clearly and in detail, we have revised the discussion section. As a result, we initially found that IVR can improve the cognitive function especially executive function and visuo-spatial attention of elderly stroke patients (P11-13 for more details).
Response 5: Please provide your response for Point 5. (in red)
- In the abstract, the authors stated that "After six weeks of treatment, the assessment scores were improved in both two groups, but there was no significant difference between the two groups" (lines 32-33). Then, in section 3.2. Results of the Cognitive Evaluation, they added:" The comparison of scores after treatment between the two groups showed that the DSST scores of the IVR group were higher than that of the control group, and the difference was statistically significant (P=0.028), but there was no significant difference in other assessment scores between the two groups" (lines 203-206). There is a contradiction in reporting the results.
Also, for statistically significant results, the effect size should be reported to show how strong the difference between the groups is.
Point 6: Abstract have made a revision: After six weeks of treatment, the assessment scores were improved in both two groups, but there was no significant difference between the two groups except DSST(Z=2.203, P=0.028<0.05).
3.2. Results of the Cognitive Evaluation (P7)have made a revision:
There were no significant differences in MOCA, TMT-A, MBI, DSST, FDST, BDST, and VFT scores between the IVR group and control group before treatment (P>0.05) (Table 2). After 6 weeks of treatment, the scores of MOCA(IVRG: T=8.981, p<0.001; CG: T=13.229, p<0.001), TMT-A(IVRG :T=5.644, p<0.001; CG: T=4.112, p=0.001), MBI(IVRG: T=-2.779, p=0.015; CG: T=-6.089, p=0.000)(Figure 2), DSST(IVGR: Z=3.422, p=0.001; CG: Z=3.482, p<0.001), FDST(IVGR: Z=2.887, p=0.004; CG: Z=2.121, p=0.034), BDST(IVGR: Z=3.317, p=0.001; CG: Z=2.111, p=0.035), and VFT(IVGR: Z=3.332, p=0.001; CG: Z=3.429, p=0.001)(Figure 3) in both groups were significantly improved compared with those before treatment(P<0.05). The comparison of scores after treatment between the two groups showed that the DSST scores of IVR group(21,6) were higher than that of control group(14,11), and the difference was statistically significant (Z=2.203, P=0.028<0.05), but there was no significant difference in other assessments scores between the two groups.
Please do not hesitate to contact me again if there is any imperfection in the revision.
Response 6: Please provide your response for Point 6. (in red)
- In the Discussion section, authors should discuss the main significant results and how they can be interpreted from the perspective of previous studies and the working hypotheses. The findings and their implications should be discussed in the broadest context possible, with references to recent research findings. Future research directions may also be highlighted.
The conclusions of the study are written ambiguously and do not respect the requirements of research with inferential objectives.
Point 7: Thanks for your suggestion, we have revised the discussion section. We focus on IVR improved executive and visuospatial functions. The possible mechanisms were discussed in combination with previous studies. Finally, we prospected the application of IVR in elderly patients with chronic stroke and the opportunity to carry out IVR in the community (P11-13 for more details).
Response 7: Please provide your response for Point 7. (in red)

Reviewer 3 Report
Abstract:
Based on the abstract the study has reported improvement in score but the improvement in score between the two groups (control and therapeutic) is not significant. This statement suggests that the IVR doesn't impact the score improvement. Therefore, either your hypothesis regarding IVR fails or there is an error in your writing. I recommend providing more details on what scores showed significant differences and what scores did not show significant differences.
Introduction
The introduction is nicely written but currently, it's not clear from the introduction, what is the goal of your study and why your study is novel compared to others. Therefore currently the introduction lacks the story of your study. You have discussed plenty of literature and its clear that IVR can be an important tool for improving cognition. I recommend breaking down the introduction into background, research gap and how your study fills it, and what is novel about it.
Methods
The level of stroke patients recruited is mild. I recommend providing or reporting more information about the measure used to determine the severity level of the score (ex, fugylmeyer score or any motor assessment test)
In the statistical analysis section, have you performed a power analysis test to determine the sample size? In addition, have you performed any Bonferroni corrections for p values and the number of tests performed? Kindly add the information.
Results
The images are nicely depicted. However, one of my concerns regarding the results is the feasibility of your IVR program. The effect of IVR can only be observed in the form of a DSST test. What is the reason other tests are not able to show a significant difference? This is something I will highlight in my results and/or discussion.
I will emphasize more on DSST more and report what cognitive improvements can be observed if your score is higher on the DSST test, and what factors other tests were unable to account for that DSST can. There must be something that makes the stroke patients perform better on the DSST test, probably the visual information is more nicely presented and easier for them to decipher. Qualitative information is hard to process compared to quantitative. This is something you can infer from the result or discussion. This will improve the quality of your manuscript.
Discussion
As mentioned in my previous comments. I would add a section where I will discuss the characteristics of different tests and try to infer why DSST scores were significantly different from the rest. What information is perceived better in this particular test as they were able to score better? How significant difference in this test and the insignificant difference in the rest of the test justify the efficacy of IVR in cognitive improvement? These are some of the questions that must be addressed in the discussion.
Conclusion
The conclusion is fine right now and should be amended once the changes in the manuscript are added, espescially about higher DSST test score.

Author Response
Response to Reviewer 3 Comments
Point 1 Abstract: Thank you for your recommendation. We provide more details in Abstract as fellows: After six weeks of treatment, the cognitive assessment scores were improved in both two groups. Moreover, the IVRG showed more improvements than the CG in the DSST(Z=2.203, P=0.028<0.05),while MOCA(T=1.186, p=0.246>0.05), TMT-A(T=1.791, p=0.084>0.05), MBI(T=0.783, p=0.44>0.05)(Figure 3), FDST(IVGR: Z=0.78, p=0.435>0.05), BDST(Z=0.347, p=0.728>0.05), and VFT(IVGR: Z=1.087, p=0.277>0.05) were not significantly improved.
Point 2 Introduction:
Background: Aging and stroke are currently major health problems in China. This problem not only affects the lives of individuals, caregivers, and families, but also places a heavy financial burden on entire medical resources.
Research gap: Most studies focus on cognitive dysfunction in the acute phase, but ignore cognitive decline in the chronic phase and training contents and scene design of IVR are relatively unitary. Currently, there are still no systematic and comprehensive cognitive training programs for IVR applications in the market.
Filling and novelty: To solve these problems, we combined IVR with puzzle games. Our IVR used a light HMD, stress-free smart sensor. The system contained 3 major categories and 16 intelligible puzzle game items for therapists and patients to choose from. To solve these problems, we combined IVR with puzzle games. Our IVR used a light HMD, stress-free smart sensor. The system contained 3 major categories and 16 intelligible puzzle game items for therapists and patients to choose from. The games interface is clear and real, bringing in a strong feeling which can "fake the real". Puzzle games can be a pattern that includes different forms and a variety of contents training programs. Recent studies indicate puzzle games are enjoyable, repeatable, easy to operate by the elderly, and can improve attention, visuospatial, and executive functions. We selected elderly patients in the chronic phase of stroke, and aim to explore the initial effectiveness, feasibility and safety of this intelligent training system in elderly patients with post-stroke cognitive impairment.
Point 3 Methods: We agree with you that movement disorders interfere with the completion of tasks, and unsuccessful actions may be the result of movement rather than cognitive deficits. Therefore, our inclusion criteria required The Fugl-Meyer motor scale>85 at least one upper and lower limb.
In our study, the PASS15 software was used to calculate the sample size based on the assumption of equal variance of two samples. We set Power was 0.80, and α was 0.05. The Montreal Cognitive assessment (MoCA) was used as the outcome index according to the previous similar research literature. We assumed that the cognitive improvement effect of the IVR group was 3.2 higher than that of the non-IVR group as effective, the standard deviation of individual MoCA was 2.2, and the loss rate was 20%. Finally, 15 cases in the experimental group and 15 cases in the control group were enrolled, and a total of 30 cases were obtained.
In our statistical analysis, we used pairwise comparisons. Specifically, the effect of IVRG before and after treatment was compared. The effects of CG before and after treatment were compared. And the effect of IVRG and CG after 6 weeks of treatment was compared. Considering that Bonferroni corrections test n independent hypotheses on the same data set this method may not be suitable for us.
Point 4 Results: Thank you for your advice. The DSST score is used to measure executive cognitive function and visuo-spatial attention. In older adults, these domains all play an important role in the cognitive tasks. The difference in DSST before and after IVR treatment suggests that VR intervention treatment could significantly improve executive and visuospatial function compared with conventional rehabilitation. The mechanism may be as follows: 1. IVR could stimulate the activation of sensory functions in brain regions related to executive function through multisensory input. 2. We combined exergames to improve executive function in older adults by enhancing presence during exercise and also by increasing patient motivation more than traditional physical activity. 3. IVR may improve the visual spatial of elderly patients through the characteristics of ecological validity and could make subjects mistakenly believe that they are in the real world through immersive stimulation. The interference from the outside world can be eliminated and the participants can be immersed in the virtual world through increased attention and reduced distraction (P12-13 for more details)
Point 5 Discussion: The DSST score represents executive cognitive function and visuo-spatial attention. Based on your valuable suggestions, we have revised the Discussion section to discuss the mechanism of why IVR can significantly improve executive cognitive function and visuo-spatial attention. Executive function plays an important role in improving the ability of daily living. But, unfortunately, the IVRG did not show a significant advantage over the non-IVRG in our results. This may be related to the standard of the MBI scale, although MBI is reliable and effective, it lacks detailed assessment of the cognitive field and the participation of some social activities. Moreover, our puzzle games are extensive but goal-oriented need for further improvement and the complexity of future life skills training projects and steps need to be further upgraded. Some studies suggest IVR could not only be used to train executive ability, spatial disorientation but also improve episodic and verbal memory, attention, living ability. However, because different training items have varying training effects, the results of our study are different from those of previous studies. Then, we have given some examples of previous VR shopping games or showcasing familiar environments to help patients improve their memory. Combined with our research, most of our training was for the patient to complete a task but did not involve memorizing and recalling. However, the performance and memory of the IVRG group were improved compared with those before treatment. The reason may be considered that VR can improve the memory function of the elderly by enhancing concentration.
Point 6 Conclusion: Thank you for your friendly reminder. We have made the following changes to the conclusions: Our research has preliminarily demonstrated that the IVR-based puzzle games may improve global cognitive, episodic memory, verbal memory, attention and daily living ability, especially in executive ability and spatial orientation in elderly patients with post-stroke cognitive impairment and this intelligent interactive experience has better applicability in the elderly group.

Round 2
Reviewer 2 Report
The authors have improved their manuscript and most of my comments have been addressed.
However, as an omission, the size effect of the differences between groups was not calculated and discussed in the Results section.
Author Response
The comparison of scores after treatment between the two groups showed that the DSST scores of IVR group(21,6) were higher than that of control group(14,11), and the difference was statistically significant (Z=2.203, P=0.028<0.05, η2=0.16), while MOCA(T=1.186, p=0.246>0.05, d=0.44), TMT-A(T=1.791, p=0.084>0.05, d=0.65), MBI(T=0.783, p=0.44>0.05, d=0.28)(Figure 3), FDST(Z=0.78, p=0.435>0.05, η2=0.02), BDST(Z=0.347, p=0.728>0.05, η2=0.004), and VFT(Z=1.087, p=0.277>0.05, η2=0.039) were not significantly improved.
The size effect of the differences between groups in DSST is η2=0.16>0.14,and P=0.028<0.05, which means the significant difference in DSST is reliable.
Reviewer 3 Report
Most of my comments have been addressed.
However, I have questions and if they are right. Would you kindly address them in your manuscript?
1. Does DSST significant difference represent an improvement in visual-spatial cognitive characteristics? It also suggests that other scores such as MBI don't show such features. Therefore, you did not observe significant differences through this other measure. I recommend adding it more clearly in your abstract and conclusion.
2. In the introduction, you mentioned your IVR puzzle game features. However, my question is why your method is unique compared to others as this method is quite well used in Rehab. In simple terms, what advantage do these features present in your methods that have not been lacking from previous literature?
Author Response
Response to Reviewer 3 Comments
Point 1: DSST significant difference represent an improvement in executive function and visual-spatial cognitive characteristics. And in other assessment scores such as DST, MBI, don't show such features. Therefore, we did not observe significant differences through this other measure. (P1, P11)
Point 2: Compared with the previous literature, our advantages are: The IVR puzzle game system contains plentiful training content, the product structure is simple, and it is easy for the elderly to understand and operate. Completion of the task is not dependent on motor function and is suitable for patients with varying degrees of limb impairment. It can also be combined with rehabilitation equipment during training, which is practical. The system has an automatic feedback system and does not rely on human supervision.